# OpenVLThinker: Complex Vision-Language Reasoning via Iterative SFT-RL Cycles

**Yihe Deng, Hritik Bansal, Fan Yin**
**Nanyun Peng, Wei Wang, Kai-Wei Chang**

University of California, Los Angeles

## Abstract

We introduce *OpenVLThinker*, one of the first open-source large vision–language models (LVLMs) to exhibit sophisticated chain-of-thought reasoning, achieving notable performance gains on challenging visual reasoning tasks. While text-based reasoning models (e.g., Deepseek R1) show promising results in text-only tasks, distilling their reasoning into LVLMs via supervised fine-tuning (SFT) often results in performance degradation due to imprecise visual grounding. Conversely, purely reinforcement learning (RL)-based methods face a large search space, hindering the emergence of reflective behaviors in smaller models (e.g., 7B LVLMs). Surprisingly, alternating between SFT and RL ultimately results in significant performance improvements after a few iterations. Our analysis reveals that the base model rarely exhibits reasoning behaviors initially, but SFT effectively surfaces these latent actions and narrows the RL search space, accelerating the development of reasoning capabilities. Each subsequent RL stage further refines the model's reasoning skills, producing higher-quality SFT data for continued self-improvement. OpenVLThinker-7B consistently advances performance across six benchmarks demanding mathematical and general reasoning, notably improving MathVista by 3.8%, EMMA by 2.4%, and HallusionBench by 1.6%. Beyond demonstrating the synergy between SFT and RL for complex reasoning tasks, our findings provide early evidence towards achieving R1-style reasoning in multimodal contexts. The code, model and data are held at https://github.com/yihedeng9/OpenVLThinker.

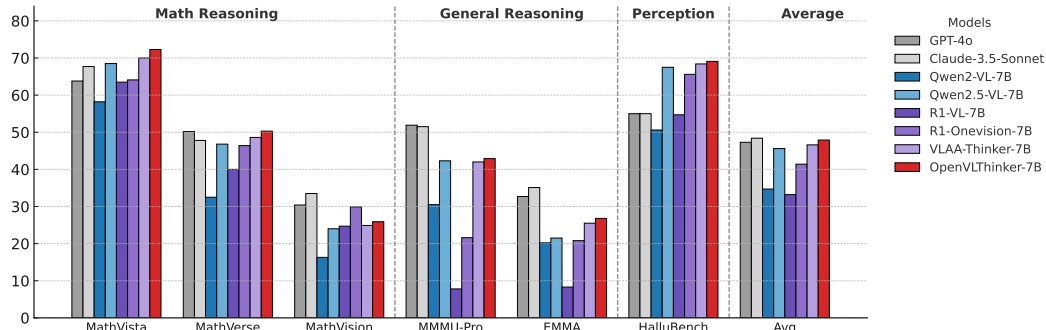

Figure 1: Our OpenVLThinker-7B (in red) performs competitively to large proprietary multimodal models such as GPT-4o and Claude-3.5 (in gray), especially in Math and Perceptron tasks. It outperforms VL base models at the same scale (in blue) and other recently released VL reasoning models (in purple).

39th Conference on Neural Information Processing Systems (NeurIPS 2025).

# 1 Introduction

Proprietary large language models (LLMs), notably OpenAI's o-series [30] and Google's Gemini-2.5 Pro [18], have demonstrated impressive multi-step reasoning abilities of planning, reflection, and verification. Recent open-weight models [53, 28, 47, 84, 83] (e.g., DeepSeek-R1 [20] and smaller LLMs like S1 [49] and QwQ-32B [63]) show that reinforcement learning (RL) with verifiable rewards effectively reproduces these advanced capabilities, significantly boosting performance on challenging mathematical and logical tasks.

Unlike text-only LLMs, it remains unclear whether open-source large vision-language models (LVLMs) can effectively adopt similar sophisticated reasoning strategies. Modern LVLMs such as LLaVA-NeXT [37] and Qwen2.5-VL [3] benefit from extensive vision-language pretraining and demonstrate strong visual instruction-following capabilities. However, they rarely demonstrate advanced reasoning behaviors like GPT-o1 or DeepSeek-R1.

Moreover, it is known that reasoning capabilities can generally be distilled from larger LLMs to smaller ones through supervised fine-tuning (SFT) on chain-of-thought demonstrations [35, 32] for text-only tasks. This recipe has been recently applied in distills demonstrations from DeepSeek-R1 (LIMO [78], S1 [49] and OpenThinker [62]) followed by optional RL fine-tuning [79]. However, adapting this method to LVLMs does not work. Proprietary LVLMs, such as OpenAI's o1/o3 and Google's Gemini, do not expose their internal reasoning paths, making their outputs unsuitable for distillation. Therefore, most recent attempts are focusing on improving LVLMs through distillation from text-only R1 reasoning models (see discussion in Section 2.2). Unfortunately, our experiments show that naively fine-tuning LVLMs on reasoning paths generated from text-based DeepSeek-R1 with image captions leads to a non-trivial performance drop (see Figure 3), primarily due to a lack of precise visual grounding. Similar observations can be found in [6, 76].

In this paper, we present OpenVLThinker-7B, one of the **first** open-weight LVLMs that exhibit complex reasoning capabilities in complex vision-language tasks. Specifically, it is trained by iterating between the following two steps:

1. *Lightweight SFT*. In the first iterations, we distill CoTs using a text-only Deep-Seek R1 given the task question and the corresponding generated image caption. These CoT traces provide demonstrations of reasoning actions, although they do not immediately improve LVLM's accuracy. For later iterations, we use the LVLM from the previous iteration to produce CoTs on 3,000 data points. This small dataset is sufficient to progressively enhance the model's reasoning depth.
2. *Curriculum RL*. In subsequent iterations, we further enhance the LVLM's reasoning through RL exploration with *Group Relative Policy Optimization* (GRPO) [56], which splits training into two rounds to form a smooth curriculum.

We found that while the initial step of SFT leads to a performance drop, iteratively alternating between SFT and RL eventually gradually yields a significant performance gain on both reasoning depth and answer accuracy (Figure 2).

Our further analysis shows that the inference-time reasoning behaviors are often triggered by specific tokens (e.g., "wait"). SFT serves as an inductive prior that highlights these reasoning actions. Specifically, it demonstrates the tokens such as "first", "wait", "check", that trigger the model's planning, reflection, and verification behaviors. Without this SFT step, launching RL from scratch forces the model to search through a prohibitively large space, making reflective behaviors slow to emerge – if they emerge at all. On the other hand, RL plays the critical role in learning the reasoning behaviors, generalizing from training data, and offering a better foundation for the next SFT iteration. The iterative cycle between SFT and RL collaboratively optimizes LVLM's performance.

We highlight our contributions as follows:

- We introduce **OpenVLThinker-7B**, one of the first open-source LVLMs to demonstrate reliable self-reflection, planning, and correction in visual contexts.
- We present a simple yet effective iterative SFT-RL loop that enables R1-style reasoning into multimodal domains and steadily self-improves without requiring massive datasets.
- We analyse linguistic markers of complex reasoning and show that SFT can steer RL exploration toward highlighted reasoning actions.
- On six challenging benchmarks, including MathVista and MathVerse, OpenVLThinker presents remarkable improvements while reducing hallucination on HallusionBench.

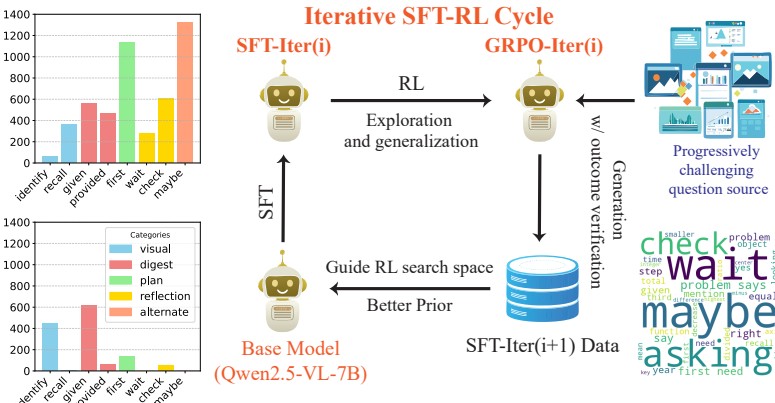

Figure 2: Illustration of OpenVLThinker-7B's training process. We iteratively apply SFT and GRPO to refine the LVLM using reasoning data generated from previous iterations. The data sources are also progressively evolved to introduce more challenging questions over time.

## 2    Related Work

### 2.1    Complex Chain-of-Thought Reasoning

Since the introduction of OpenAI's O1 model [30], researchers have shown strong interest in reproducing and enhancing the complex reasoning capabilities of LLMs [53, 28, 47, 84, 83], partly due to its superior performance on mathematical benchmarks. [20] introduce the open-source *DeepSeek-R1* model and investigate how RL with verifiable rewards can promote advanced chain-of-thought reasoning and reflective behaviors. This development inspired a line of research focused on open-source reproduction [45, 25, 85, 41, 62] and the analysis of such complex reasoning in mathematical problem solving [79, 73, 78, 10]. In parallel, several recent studies have similarly explored the effects of test-time scaling on encouraging more complex model reasoning behaviors [49, 55, 39, 17, 87, 60]. However, the majority of research have significantly advanced text-based reasoning, and development of vision-language reasoning is much more initial.

### 2.2    Vision-Language Reasoning Model

Recent advancements in large vision-language models (LVLMs) stem from open-source LLMs [65, 66, 14, 75] and text-aligned image encoders [54, 36]. Integrating these components has enabled LVLMs to follow diverse visual instructions and generate meaningful responses [38, 15, 16, 11, 37, 3]. Parallel to the model development, researchers have also been interested in eliciting CoT reasoning chains from LVLMs via prompting [89, 90, 48, 26] or fine-tuning [21, 74, 64, 13]. These reasoning models remain mostly on a shallow level of common step-by-step prompting, without self-reflections or self-verifications.

**Concurrent work.** Very recently, many studies have started exploring how to equip LVLMs with R1-like reasoning capabilities through distillation from text-only reasoning models [5, 76, 27, 33] or directly rely on RL [91, 43] for self-exploration. Further advancements [57, 71, 6, 46, 72, 68, 40, 86] have focused on improving performance in visual math reasoning, which marks the transition from early-stage exploration to more effective complex vision-language reasoning. Please note that most of these works are within the two months before the submission date, and some of them do not even have associated technical reports available yet. Our work aligns with these studies and contributes unique insights into the role of SFT for complex reasoning, along with an iterative SFT-RL framework to further advance research in this direction.

## 3    Preliminaries

An LLM is defined by a probability distribution $p_{\boldsymbol{\theta}}$, parameterized by model weights $\boldsymbol{\theta}$. Given a prompt sequence $\mathbf{x} = [x_1, \ldots, x_n]$, the model generates a response sequence $\mathbf{y} = [y_1, \ldots, y_m]$, where $x_i$ and $y_j$ represent individual tokens. The response $\mathbf{y}$ is sampled from the conditional distribution $p_{\boldsymbol{\theta}}(\cdot|\mathbf{x})$, factorized as $p_{\boldsymbol{\theta}}(\mathbf{y}|\mathbf{x}) = \prod_{j=1}^{m} p_{\boldsymbol{\theta}}(y_j|\mathbf{x}, y_1, \ldots, y_{j-1})$.

**Supervised Fine-Tuning (SFT).** SFT is typically applied to specialize LLMs for a particular task or domain. This process updates the model parameters $\boldsymbol{\theta}$ by providing example responses of desired

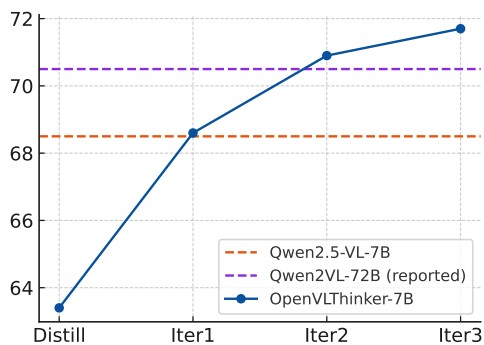

Figure 3: Iterative performance improvement of our model on MathVista. We note that *Iter(i)* is always fine-tuned from the base model Qwen2.5-VL-7B, with its training data generated from *Iter(i-1)*.

Figure 4: An example of OpenVLThinker-7B reasoning. Question: In the diagram of the food web shown, if the number of ferns decrease, the supply of salmon will most likely? (A) decrease (B) can't tell (C) stay same (D) increase. Corresponding image is shown in Figure 12.

behavior to the input instructions. Concretely, Given a dataset $\mathcal{D} = \{(\mathbf{x}^{(i)}, \mathbf{y}^{(i)})\}_{i=1}^{N}$, where $\mathbf{x}^{(i)}$ is the prompt sequence and $\mathbf{y}^{(i)}$ is the desired response sequence. We update $\boldsymbol{\theta}$ to maximize the likelihood of producing $\mathbf{y}^{(i)}$ given $\mathbf{x}^{(i)}$. Formally, $\mathcal{L}_{\mathrm{SFT}}(\boldsymbol{\theta}) = -\sum_{i=1}^{N} \log p_{\boldsymbol{\theta}}(\mathbf{y}^{(i)} \mid \mathbf{x}^{(i)})$. By minimizing the loss, the model learns to produce responses more aligned with the labeled examples.

**Reinforcement Learning (RL).** RL approaches fine-tune LLMs via human preferences modeled under the Bradley-Terry model [50, 12, 56, 1]: $p(\mathbf{y}_w \succ \mathbf{y}_l \mid \mathbf{x}) = \sigma(r(\mathbf{x}, \mathbf{y}_w) - r(\mathbf{x}, \mathbf{y}_l))$, where $\mathbf{y}_w$ and $\mathbf{y}_l$ denote preferred and dispreferred responses, respectively, and $\sigma(t) = 1/(1 + e^{-t})$ is the sigmoid function. The common RL objective under the Bradley-Terry assumption of the reward model $r(\mathbf{x}, \mathbf{y})$ is thus

$$\max_{\boldsymbol{\theta}} \left[ \mathbb{E}_{\mathbf{x}, \mathbf{y} \sim p_{\boldsymbol{\theta}}}[r(\mathbf{x}, \mathbf{y})] - \beta \, \mathbb{E}_{\mathbf{x}} \left[ \mathrm{KL}(p_{\boldsymbol{\theta}}(\cdot|\mathbf{x}) \| p_{\mathrm{ref}}(\cdot|\mathbf{x})) \right] \right],$$

where $\beta > 0$ is the KL penalty coefficient. Under this framework, [56] introduced Group Relative Policy Optimization (GRPO) by sampling a group of response trajectories $\{\mathbf{o}_i\}_{i=1}^{G}$ from the old policy model $\boldsymbol{\theta}_{\mathrm{old}}$ for each query $\mathbf{x}$, with the objective as maximizing:

$$\mathbb{E}\left[ \frac{1}{G} \sum_{i=1}^{G} \frac{1}{|\mathbf{o}_i|} \sum_{t=1}^{|\mathbf{o}_i|} \min\left( \frac{p_{\boldsymbol{\theta}}(o_{i,t} \mid \mathbf{x}, \mathbf{o}_{i,<t})}{p_{\boldsymbol{\theta}_{\mathrm{old}}}(o_{i,t} \mid \mathbf{x}, \mathbf{o}_{i,<t})} \widehat{A}_{i,t}, \mathrm{clip}\left( \frac{p_{\boldsymbol{\theta}}(o_{i,t} \mid \mathbf{x}, \mathbf{o}_{i,<t})}{p_{\boldsymbol{\theta}_{\mathrm{old}}}(o_{i,t} \mid \mathbf{x}, \mathbf{o}_{i,<t})}, 1-\epsilon, 1+\epsilon \right) \widehat{A}_{i,t} \right) \right]$$

$$- \beta \, \mathbb{D}_{\mathrm{KL}}\left[ p_{\boldsymbol{\theta}} \, \| \, p_{\boldsymbol{\theta}_{\mathrm{ref}}} \right], \tag{1}$$

where $\epsilon > 0$ is a hyperparameter bounding the clipping range, $\beta > 0$ balances the KL-penalty term $\mathbb{D}_{\mathrm{KL}}[\pi_\theta \, \| \, \pi_{\mathrm{ref}}]$ against the advantage-weighted policy update, and $\boldsymbol{\theta}_{\mathrm{old}}$ is the old policy model. Here, the advantage $\widehat{A}_{i,t} = \widetilde{r}_i = (r_i - \mathrm{mean}(r))/\mathrm{std}(r)$ is set as the normalized reward at group level.

## 4 OpenVLThinker: Iterative Self-improvement on Curriculum Data

In this section, we first analyze how SFT and RL affect the occurrence of reasoning-related keywords, which serves to motivate our approach. We then introduce the proposed iterative approach to enhancing complex reasoning capabilities in OpenVLThinker-7B with SFT-RL cycles. At last, we propose a source-based curriculum RL.

### 4.1 The Role of SFT and RL

**The initial SFT data.** The standard distillation approach used for text-only reasoning cannot be directly applied because the R1 model does not support visual input, and other proprietary LVLMs, such as OpenAI's o1/o3, do not expose their internal reasoning paths. To learn reasoning behaviors from R1, we instead use the target model as a captioning model, prompting it to generate detailed textual descriptions for each image. Subsequently, these captions serve as proxies for the images when input into a text-based R1 reasoning model, *QwQ-32B* [63], which then generates $k$ candidate reasoning chains. Among these candidates, we select the shortest reasoning chain that correctly

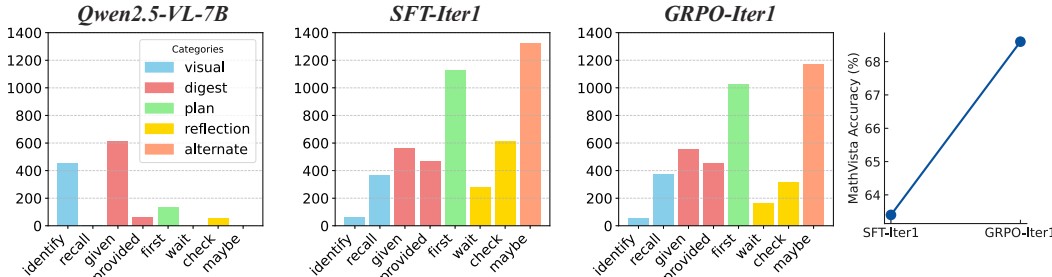

Figure 6: Occurrences of reasoning keywords when solving MathVista with the base model, SFT-Iter1 model, and GRPO-Iter1 model. The most significant distribution shift occurs after SFT, while the scale remains largely unchanged after GRPO, despite notable performance improvements.

arrives at the final answer to avoid excessive reasoning length after SFT (further details in Section 5.2). The overall procedure is summarized in Figure 5.

**Impact of SFT and RL on Model Reasoning Actions.** Complex reasoning behaviors in LLMs have been described using various terms, including long CoT [79] and aha moments [20]. At their core, these behaviors reflect autonomous planning, reflection, and verification steps that occur during inference. We refer to them as *inference-time actions*, which are often triggered by specific tokens such as "wait". To examine how SFT and RL influence these reasoning actions, we identify eight representative keywords corresponding to perception, question comprehension, planning, reflection, and seeking alternatives.

As illustrated in Figure 6, the base model seldom exhibits planning, reflection, or alternative-solution actions. However, SFT

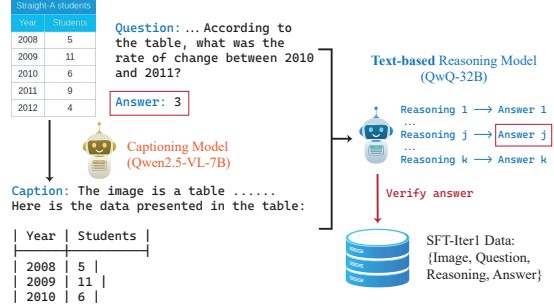

Figure 5: Curation of SFT-Iter1 data from text-based reasoning models based on image descriptions.

guided by text-based R1-like reasoning traces effectively surfaces these behaviors. As shown in the third and fourth subplots of Figure 6, subsequent GRPO-based RL training following SFT-Iter1 substantially enhances model performance on MathVista by 5.2%, yet largely maintains the initial reasoning action distribution, with minor refinements such as reduced repetitive reflections.

Conversely, direct RL training without prior SFT struggles to efficiently induce reasoning behaviors, exemplified by the absence of reflection keywords (e.g., "wait") even after an equivalent training volume. Concurrent research by [68], which solely relies on RL, addresses this by explicitly appending relevant keywords during training rollouts. These observations support our argument that SFT plays a critical role in highlighting desirable reasoning actions, providing an efficient and effective foundation for RL to build upon. In contrast, RL primarily serves to further refine and enhance performance.

### 4.2 Iterative Improvement

The model obtained after the first iteration (GRPO-Iter1) demonstrates enhanced complex reasoning capabilities and improved reliability in processing visual inputs compared to methods based on image-to-text conversion. This advancement positions GRPO-Iter1 as an effective source for generating higher-quality reasoning demonstrations. Consequently, we propose an iterative self-improvement strategy, inspired by established methodologies such as iterative SFT in ReST-EM [59] and iterative direct preference optimization (DPO) schemes [81, 51], both of which have shown substantial effectiveness in iterative training processes and fall under the Expectation-Maximization framework [59].

Specifically, in each iteration, we sample a new set of enhanced reasoning traces using the model trained in the preceding iteration. These refined demonstrations are then utilized to retrain the base

model[1], thereby progressively elevating its reasoning performance. The overall iterative pipeline is illustrated in Figure 2, and the consistent incremental performance gains achieved through successive iterations are depicted in Figure 3.

## 4.3 Two-Stage Source-Based Curriculum RL

To ensure effective exploration during reinforcement learning (RL), we assess the difficulty of data sources, aiming to provide data that is challenging yet appropriate for the model's proficiency level. Specifically, we utilize GPT-4o to rate the difficulty of five representative examples drawn from various data sources such as FigureQA [31], MapQA [4], and GeoQA [7], in a similar fashion to the text-based evaluation in DeepMath-103K [23]. Additionally, we employ the base model, Qwen2.5-VL-7B[2], to obtain its error rates as a complementary difficulty indicator. We standardize independently using z-score normalization for both the GPT-4o rating and base model error rates and compute the average of the two. Based on this composite score, we categorize the data sources into Easy, Medium, and Hard groups via k-means clustering in 1d space. With these categories, we construct two difficulty-specific datasets: $\mathcal{D}_{\mathrm{RL(Medium)}}$ and $\mathcal{D}_{\mathrm{RL(Hard)}}$. Our curriculum training thus proceeds in two stages within one iteration, sequentially training on $\mathcal{D}_{\mathrm{RL(Medium)}}$ and $\mathcal{D}_{\mathrm{RL(Hard)}}$.

## 5 Experiments

**Training setup.** We take Qwen2.5-VL-7B [3] as the base model and perform three iterations of the SFT-RL cycle as illustrated in Section 4, applying full fine-tuning for both SFT and RL. Our training framework is based on LLaMA-Factory[3] for SFT and EasyR1[4] for RL. We source our training data from the established LLaVA-OneVision [34] and specifically consider the 14 data sources in overlap with MathV360K [58] (Table 4). Based on our preliminary experiments, we equally draw 500 examples from each source to form the SFT seed dataset of 7K examples, where for each iteration we collect distillation data via rejection sampling, resulting in a final 3K SFT data. We then classify the data sources into easy, medium and hard (as detailed in Table 4). We construct the 3K medium-level RL training data from the 5 sources that we identified as medium difficulty. Finally, we construct 6K hard-level RL training data from the 3 most difficult sources, summing up to 12K data in total for each iteration that trains from the base model. We defer the training hyperparameters to Appendix C.

**Evaluation.** Our evaluation employs exact matching and a grader function from MathRuler[5]. We use the same inference hyperparameter as suggested by Qwen and recovered Qwen2.5-VL-7B's reported results on MathVista at 68.5%. The hyperparameters are detailed in Table 12. We employ six established benchmarks to examine model's ability thoroughly:

- Math reasoning: MathVista [44], MathVerse [88] and MathVision [69]. The three benchmarks evaluate how LVLMs interpret and reason with diagrams in visual math problems through both multiple-choice and free-form questions.
- General reasoning: MMMU-Pro [82] and EMMA [22]. MMMU-Pro spans 30 subjects across 183 subfields, including business, medicine, and science. EMMA evaluates in physics, chemistry, coding, and math.
- Perception: HallusionBench [19], designed to evaluate LVLMs' susceptibility to language hallucination and visual illusion.

**Baselines.** We evaluate the non-reasoning base model Qwen2.5-VL-7B as a primary baseline to demonstrate the improvements introduced by our method. Additionally, we include the reported performance of proprietary models, including GPT-4o [29] and Claude-3.5-Sonnet [2], alongside open-source LVLMs such as Mulberry-7B [77], InternVL2.5-8B [9], Kimi-VL-16B [61], and Qwen2-VL-7B [70], as reference points. Crucially, to highlight the effectiveness of our iterative SFT-RL training strategy, we compare our model with concurrent approaches employing a single round of SFT distillation and RL at the same model scale (7B), yet utilizing significantly larger training datasets. These concurrent models include R1-VL-7B [86], R1-Onevision-7B [76], and VLAA-Thinker-Qwen2.5VL-7B [6]. Notably, R1-Onevision and VLAA-Thinker-Qwen2.5VL-7B also start from the same base model (Qwen2.5-VL-7B) as ours, using 165K and 150K total data, respectively.

---

[1]To maintain stability, we retrain the model from scratch at each iteration with the newly generated dataset, as similar to some iterative approaches in text-only domain [59, 24].

[2]In alignment with previous R1 reasoning research [79, 78], we choose the base model from Qwen2.5 family for their strong general capability obtained in pre-training.

[3]https://github.com/hiyouga/LLaMA-Factory

[4]https://github.com/hiyouga/EasyR1

[5]https://github.com/hiyouga/MathRuler

Table 1: Evaluation results across visual math reasoning benchmarks (MathVista, MathVerse, MathVision), general visual reasoning benchmarks (MMMU-Pro, EMMA), and perception (HallusionBench). We include the reported performance of proprietary models and open-source Vision-Language models as references. *Performance of the base model Qwen2.5-VL-7B and concurrent reasoning models are evaluated by us under the same setting and hardware as OpenVLThinker. The **bold** numbers indicate the best results among the open-source models and the underscored numbers represent the second-best results.

| Model | Data | Math Reasoning | | | General Reasoning | | Visual | Avg |
| | | Math-Vista | Math-Verse | Math-Vision | MMMU-Pro | EMMA | Hallu-Bench | |
|---|---|---|---|---|---|---|---|---|
| *Proprietary Model* | | | | | | | | |
| GPT-4o | - | 63.8 | 50.2 | 30.4 | 51.9 | 32.7 | 55.0 | 47.3 |
| Claude-3.5-Sonnet | - | 67.7 | 47.8 | 33.5 | 51.5 | 35.1 | 55.0 | 48.4 |
| *Open-source Vision-Language Model* | | | | | | | | |
| Mulberry-7B | - | 63.1 | 39.6 | - | - | - | 54.1 | - |
| InternVL2.5-8B | - | 64.4 | 39.5 | 19.7 | 34.3 | - | - | - |
| Kimi-VL-16B | - | 68.7 | 44.9 | 21.4 | - | - | - | - |
| Qwen2-VL-7B | - | 58.2 | 32.5 | 16.3 | 30.5 | 20.2 | 50.6 | 34.7 |
| Qwen2.5-VL-7B* | - | 68.5 | 46.8 | 24.0 | 42.3 | 24.4 | 67.5 | 45.6 |
| *Concurrent Vision-Language Reasoning Models* | | | | | | | | |
| R1-VL-7B | 270K | 63.5 | 40.0 | 24.7 | 7.8 | 8.3 | 54.7 | 33.2 |
| R1-Onevision-7B | 165K | 64.1 | 46.4 | **29.9** | 21.6 | 20.8 | 65.6 | 41.4 |
| VLAA-Thinker-7B* | 150K | 70.0 | 48.6 | 24.9 | 42.0 | 25.5 | 68.4 | 46.6 |
| OpenVLThinker-7B* | 12K | **72.3** | **50.3** | 25.9 | **42.9** | **26.8** | **69.1** | **47.9** |

In contrast, our model achieves better performance with only 12K training samples from the base model.

## 5.1 Main Results

We present our main results in Figure 1, with detailed performance across datasets shown in Table 1. As illustrated, OpenVLThinker-7B consistently achieves either the best or second-best scores among open-source LVLMs of comparable scale across all six benchmarks, including concurrent reasoning models. On average, OpenVLThinker attains an accuracy of 46.6%, representing a 2% improvement over the base model and performance comparable to proprietary models such as GPT-4o. Notably, OpenVLThinker exhibits fewer hallucinations and more precise perception than its base

Table 2: Performance of 3B models on MathVista.

| Model | Accuracy (%) |
|---|---|
| R1-VL-2B | 52.1 |
| InternVL2.5-4B | 60.5 |
| Qwen2.5-VL-3B | 62.3 |
| VLAA-Thinker-3B | 61.0 |
| OpenVLThinker-3B | **63.4** |

model on HallusionBench, improving accuracy by 2.7%. Compared to concurrent reasoning methods that utilize substantially larger datasets for single-iteration SFT and RL, our iterative approach achieves superior results while utilizing only 1/10 of the data scale as used in concurrent works with a single-iteration SFT-RL pipeline.

**OpenVLThinker-3B.** We additionally train a 3B model using a single iteration of the SFT-RL pipeline, where the training process distills from our 7B model. In Table 2, we compare the performance of our 3B model against current representative models at the same scale, including our base model, Qwen2.5-VL-3B, and the reasoning model VLAA-Thinker-3B, which is trained from the same initial checkpoint as ours. OpenVLThinker-3B achieves the best performance on MathVista and outperforms state-of-the-art 3B reasoning models.

## 5.2 Analysis

**Distillation at iteration 1.** At SFT-Iter1, we utilized the base model Qwen2.5-VL-7B to generate image descriptions and obtained R1-like reasoning from QwQ-32B through rejection sampling. A

Table 3: Performance on the MathVista benchmark comparing different SFT data-filtering strategies. Removing the most repetitive keywords in data can mitigate repetitive reflections after SFT.

| Model Variant | Accuracy (%) |
|---|---|
| Qwen2.5-VL-7B | 68.5 |
| Vanilla | 57.5 |
| Filtered | 58.7 |
| Truncated (**SFT-Iter1**) | 63.4 |

Table 4: Categorization of data sources by composite difficulty score using k-means with k=3. The geometry question sources all fall into the hard category.

| Easy | Medium | Hard |
|---|---|---|
| ChartQA | FigureQA | UniGeo |
| IconQA | CLEVR | GEOS |
| VizWiz | A-OKVQA | Geometry3K |
| TabMWP | SuperCLEVR | GeoQA |
| DVQA | MapQA | |

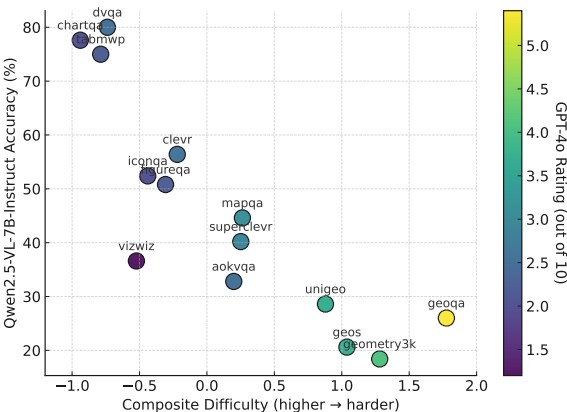

Figure 7: Data source difficulty based on base model accuracy and GPT-4o rating.

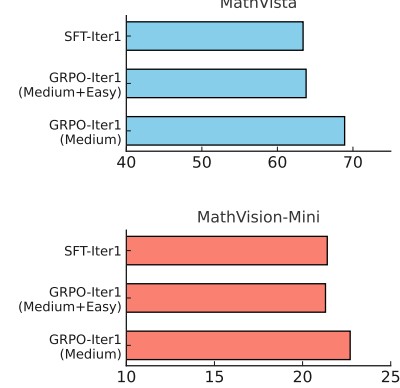

Figure 8: Performance at GRPO-Iter1 using data from different difficulty sources, at the same scale of 3K.

common problem for distillation observed in text-only math reasoning is the overly long reasoning length coupled with unnecessary repetitions of reflections [79, 42]. We observed similarly that these initial reasoning traces were often excessively verbose, partly due to information loss during image-to-caption conversion. Consequently, post-SFT reasoning became increasingly repetitive with unproductive self-reflections (see Appendix D for an illustration). To address this, we evaluated two filtering strategies: (1) discarding samples with reasoning traces exceeding 500 words, and (2) truncating reflections by splitting traces of at specific keywords that were overly repetitive in data ( "Wait," "But wait," and "But the question") and discarding subsequent segments while preserving the final answer. The latter approach was ultimately adopted to prevent the model from internalizing reflection loops, while preserving the reasoning action at a reasonable frequency. Table 3 compares models trained on original versus processed data.

**Data source difficulty.**    We conducted a quantitative analysis to categorize the data sources based on difficulty. Applying k-means clustering (with $k = 3$) to our composite difficulty score as described in Section 4 allowed us to clearly identify three distinct difficulty pools, as shown in Table 4. We visualize the difficulty scores for each source in Figure 7. In Figure 8, we show the performance of GRPO-Iter1 when drawing 3K data from either (1) 10 data sources classified as either Easy or Medium, or (2) 5 data sources classified as Medium. We observe that RL training with easy-level data results in ineffective performance gain as compared to sourcing from medium-level data only. This finding aligns with concurrent algorithmic efforts such as DAPO [80] in the text-only domain for improving GRPO by dynamically filtering out overly-easy examples.

**Curriculum RL to maximize utilization of challenging data.**    Figure 9 investigates the impact of incorporating challenging training data (e.g., geometry datasets) at iteration 1. On the left panel, we illustrate the absolute performance gains transitioning from SFT-Iter1 to GRPO-Iter1 (medium difficulty), and subsequently from GRPO-Iter1 (medium) to GRPO-Iter1 (hard). Training on these harder datasets yields substantial improvements on more difficult benchmarks, such as MathVision, while not significantly affecting performance on easier benchmarks like MathVista. On the right panel, we further compare our two-stage, source-based curriculum RL approach against training solely on

hard data. The results indicate that initiating RL with moderately challenging (medium difficulty) data and subsequently progressing to harder datasets provides optimal performance improvements.

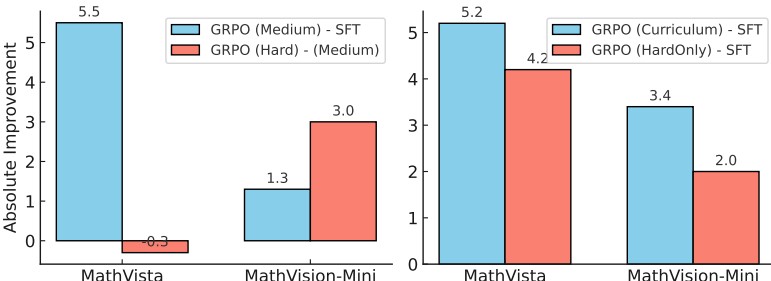

Figure 9: Absolute performance gain achieved at iteration 1. The round 2 RL training on hard data provides more significant performance gain on harder benchmarks such as MathVision. Moreover, if RL training with the hard data only yield less improvement than our two-stage curriculum RL.

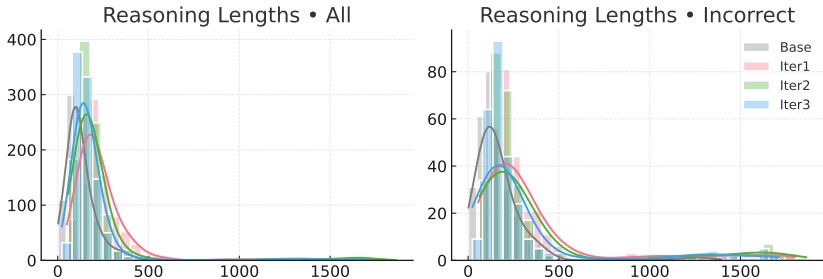

Figure 10: Distribution of reasoning length (number of words) across iterations of training. While our trained reasoning model across iterations all tend to reason longer than the base model, iterative training resulted in gradually more concise length, possibly due to reduced repetitive reflections.

**Iterative progression.** Building upon the performance improvements shown in Figure 3, we further analyze changes in reasoning length across iterations, as illustrated in Figure 10. Our results indicate that the reasoning model consistently utilizes more words at inference time compared to the base non-reasoning model, without becoming excessively repetitive. Notably, the largest increase in reasoning length occurs at Iteration 1, with subsequent iterations gradually adopting more concise reasoning. This progression suggests an increasingly efficient utilization of reflective reasoning, engaging reflections primarily when beneficial. In Appendix D (Figure 13 and 14), we show reasoning examples that our SFT-ed model was incorrect while our RL-ed model was correct.

**Design choice on restarting iterations** Restarting training from scratch at each iteration is a standard practice in iterative self-improvement methods [59, 24]. This design choice ensures training stability and prevents overfitting, especially when the data scale is relatively small, thus maintaining better generalization to unseen tasks. As noted in [59], re-training from the base model provides comparable task-specific performance and significantly better transfer to held-out tasks compared to continued training.

In our iterative re-training approach, the SFT data is refined and improved across iterations, while the base model parameters are reinitialized to prevent error accumulation. To further substantiate this design choice, we conducted an additional comparison between (a) re-training from scratch and (b) continuing training from the previous checkpoint.

The results indicate that continued training leads to performance degradation on HallusionBench, suggesting potential overfitting to the previous iteration's data. Hence, restarting from the base model offers a more robust and generalizable learning trajectory across iterations.

**Additional evaluation benchmarks.** Our main paper evaluates OpenVLThinker across six widely-used vision-language benchmarks covering mathematical reasoning, general reasoning, and perceptual

Table 5: Comparison between re-training from scratch and continued training.

| Method | MathVista | EMMA | HallusionBench |
|---|---|---|---|
| OpenVLThinker (re-training) | 71.7 | 25.8 | 70.2 |
| OpenVLThinker (continue training) | 71.8 | 25.1 | 66.8 |

reliability. These benchmarks are consistent with those used in recent reports on both proprietary (e.g., GPT, Gemini) and open-source (e.g., Qwen-VL, Intern-VL) models. This setup aligns with concurrent works [27, 6] that also employ these benchmarks with emphasis on reasoning ability.

To further clarify benchmark coverage, Table 6 provides subset-level EMMA results, showing performance across Math, Chemistry, Physics, and Code categories.

Table 6: Subset performance on EMMA benchmark.

| Model | EMMA-Math | EMMA-Chemistry | EMMA-Physics | EMMA-Code |
|---|---|---|---|---|
| Qwen2.5-VL | 24.6 | 21.9 | 29.5 | 28.0 |
| VLAA-Thinker | 28.1 | 22.3 | 28.8 | 27.3 |
| OpenVLThinker | 28.8 | 22.6 | 32.7 | 26.8 |

In addition, we expanded the evaluation to include two recent benchmarks, MM-Star [8] and We-Math [52]. MM-Star assesses six major LVLM capabilities, including fine-grained perception, mathematics, science & technology, and logical reasoning.

Table 7: Results on newly included benchmarks MM-Star and WeMath.

| Model | MM-Star | WeMath |
|---|---|---|
| Qwen2.5-VL | 53.9 | 61.9 |
| VLAA-Thinker | 55.4 | 62.4 |
| OpenVLThinker | 61.9 | 64.1 |

Together, these expanded evaluations across eight comprehensive benchmarks demonstrate the robustness and generalizability of our approach across multiple reasoning domains.

## 6   Conclusion

In this work, we proposed a new perspective on LLM reasoning as actions at inference time, signified by keywords such as "wait". We thus interpret the roles of SFT as action highlighting that efficiently surfaces desired actions by distilling a reasoning model's demonstrations. On the other hand, RL makes improvement on basis provided by SFT. Based on this intuition, we introduced OpenVLThinker-7B, a LVLM enhanced through an iterative self-improving process combining SFT and RL to enable complex CoT reasoning. Our results demonstrate that integrating R1-style reasoning into LVLMs effectively boosts their multimodal reasoning performance across benchmarks. With only three SFT-RL cycles and 12K training examples, the model raises average accuracy on six diverse visual-reasoning benchmarks to 46.6 %, with a 2 % absolute gain over its base model and on par with proprietary systems such as GPT-4o.

**Limitations.** Our experiments span six established benchmarks, yet they do not exhaustively probe robustness in other tasks or real-world settings. In addition, we validated the method only on a 7B model as a proof of concept. While the approach should scale to larger backbones (e.g., 32B) and likely yield further gains, such exploration requires substantially greater computational resources.

## Acknowledgments

We thank anonymous reviewers for their helpful comments. This work was partially supported by U.S. DARPA ECOLE Program No. #HR00112390060, ONR grant N00014-23-1-2780, DARPA ANSR program FA8750- 23-2-0004, Amazon, and Apple. Chang and Peng were supported in part by a grant from DARPA to the Simons Institute for the Theory of Computing. Bansal was supported in part by AFOSR MURI grant FA9550-22-1-0380. Wang was supported by National Science Foundation (2106859, 2200274, 2312501), National Institutes of Health (U54HG012517, U24DK097771, U54OD036472), NEC, and Optum AI.

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

# A   Additional Experiments

## A.1   Computational Cost of the Iterative SFT–RL Loop

We provide a detailed breakdown of the computational cost for each stage of the iterative SFT→RL training process. The experiments were conducted on an $8\times$H100 (or equivalent) GPU node.

Table 8: Approximate GPU hours per stage for iterative SFT→RL training.

| Stage | GPU Hours (per GPU) | Total (8 GPUs) |
|---|---|---|
| SFT | 0.06 (3 min 30 s) | 0.48 |
| GRPO-Medium | 2.01 (2 h 35 s) | 16.08 |
| GRPO-Hard | 4.57 (4 h 34 min 26 s) | 36.56 |

The SFT stage incurs minimal computational cost due to the small dataset size (3k examples). For RL training on medium-difficulty data, the number of epochs is reduced to maintain efficiency. Although the hard-stage RL incurs the highest cost, the overall compute remains comparable to contemporary RL-based post-training methods.

Importantly, the preceding SFT and medium RL stages accelerate convergence during the final RL stage. Despite the iterative nature introducing additional overhead, the total compute remains practical and resource-efficient for academic-scale training. We plan to include precise GPU-hour estimates and discuss scalability trade-offs in future versions.

## A.2   Single-Stage SFT-Only and RL-Only Baselines

To isolate the effect of iteration beyond simply combining SFT and RL, we conducted experiments with single-stage SFT-only and RL-only baselines trained on the same 12K examples used in OpenVLThinker.

- **RL-only:** Qwen2.5-VL trained exclusively with GRPO on the full 12K dataset.
- **SFT-only:** Qwen2.5-VL trained solely via SFT on iteration-2 trajectories generated by OpenVLThinker. Instances with no correct reasoning within $k = 4$ samplings were filtered.

Both baselines were trained to full convergence, and checkpoints were selected based on validation performance. OpenVLThinker was trained using the same initialization but followed the iterative SFT→RL loop.

Table 9: Comparison of single-stage baselines and iterative OpenVLThinker.

| Method | MathVista | EMMA | HallusionBench |
|---|---|---|---|
| RL-Only | 71.3 | 24.5 | 66.8 |
| SFT-Only | 71.1 | 22.3 | 65.4 |
| OpenVLThinker (Iterative) | 71.7 | 25.2 | 70.2 |

Training the RL-only baseline on the full 12K dataset required approximately 16 GPU-hours using an $8\times$H100 node—comparable to the cumulative training time of OpenVLThinker. While RL-only surpassed SFT-only, the iterative OpenVLThinker consistently achieved the best performance, demonstrating that the improvement stems from iterative refinement rather than merely combining SFT and RL.

## A.3   Impact of Caption Quality on Iterative Training

The quality of caption-based SFT data significantly influences reasoning performance in the iterative training loop. During iteration 1, we constructed the dataset using captions generated by QwQ-32B, filtered by final-answer correctness. Higher-quality captions increased the likelihood of correct reasoning traces, thus expanding the effective training pool.

To analyze this effect, we compared two variants: one using captions from the weaker Qwen2.5-VL-3B model and another using Qwen2.5-VL-7B, both under identical rejection-sampling conditions ($k = 4$). The performance evolution across iterations on the MathVista benchmark is shown below.

Better caption quality improves visual grounding and reasoning trace precision in early stages, yielding higher-quality data for subsequent iterations. Consequently, richer initial captions amplify

Table 10: Effect of caption quality across training iterations on MathVista.

| Caption Source | SFT-Iter1 | GRPO-Iter1 | SFT-Iter2 | GRPO-Iter2 | SFT-Iter3 | GRPO-Iter3 |
|---|---|---|---|---|---|---|
| 3B Caption | 62.5 | 65.6 | 66.1 | 69.4 | 69.0 | 70.2 |
| 7B Caption | 63.4 | 66.6 | 67.5 | 70.9 | 69.5 | 71.7 |

the benefits of the iterative framework, leading to more consistent improvements in reasoning performance.

# B   Additional Empirical Study

**Does Complex Reasoning Matter for VQA?**

We additionally investigated whether complex, multi-step reasoning provides significant performance gains over standard (non-R1) reasoning in visual tasks. In this study, we use the *ConTextual* [67] validation set of 100 VQA examples, aiming to disentangle the roles of image grounding and textual reasoning. As similar to our first distillation process, we separately employ a vision-language model for caption generation and a pure-text model for reasoning. The image description generated by the captioning model is then fed into one of two text-based models: *DeepSeek-R1-Distill-14B* (an R1-style reasoner) or *Qwen2.5-14B-Instruct* (a standard instruction-tuned model). This setup allows us to isolate the impact of R1 reasoning from the effects of the underlying vision encoder.

We further explore how different levels of caption quality influence final accuracy by comparing two caption generators, *LLaVA-v1.6-34B* and *GPT-4o*. Additionally, we vary the number of sampled reasoning paths ($k = 1, 2, 4$) and compute pass@$k$ accuracies for each condition. As a baseline, we include direct QA outputs from *LLaVA-v1.6-34B* without any intermediate text description (i.e., the model sees images directly). Figure 11 summarizes these results. In our experiments, we find that R1-style reasoning provides consistent benefits:

**(1) R1 reasoning outperforms standard methods.** When provided with identical captioned inputs, *DeepSeek-R1-Distill-14B* achieves higher accuracy than *Qwen2.5-14B-Instruct*. Moreover, its performance can match (or even surpass) the direct QA accuracy of its own captioning model (*LLaVA-v1.6-34B*), despite potential information loss from translating the image into text.

**(2) Sampling benefits complex reasoners.** Increasing the number of sampled reasoning chains ($k = 2$ or $k = 4$) leads to larger performance gains for R1 models than for standard Qwen models, indicating that the multi-step reasoning approach can more effectively converge on correct solutions when multiple hypotheses are explored.

**(3) Image grounding quality matters.** We observe that richer and more precise captions significantly enhance final VQA accuracy. When captions are more detailed (e.g., from *GPT-4o*), the improvements from complex reasoning are especially pronounced.

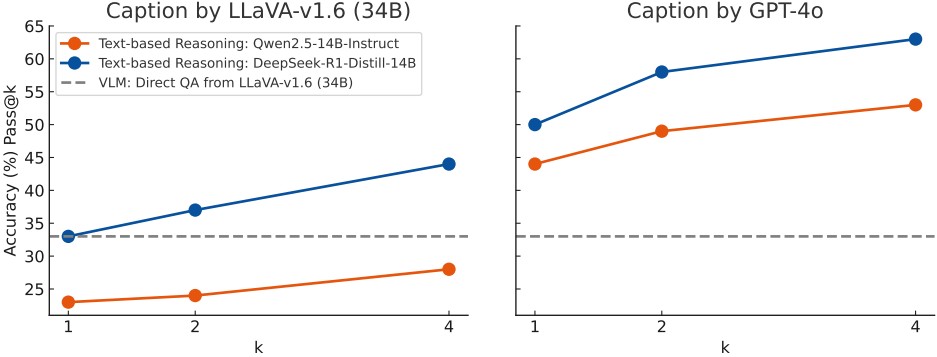

Figure 11: Pass@k accuracy of different reasoning models based on captions generated with different vision-language LLMs.

Table 11: VQA accuracy after a single round of caption refinement. While pass@4 increases slightly, pass@1 and pass@2 remain largely unchanged.

| Caption Type | pass@1 | pass@2 | pass@4 |
|---|---|---|---|
| Original | **33** | **37** | 44 |
| Refined | 29 | 35 | **46** |

**Caption Refinement via Feedback: Limited Effectiveness.**

We also investigated whether a one-round feedback loop could improve the quality of captions and thus final VQA performance. Concretely, *DeepSeek-R1-Distill-14B* was prompted to list missing or ambiguous details in the initial captions generated by *LLaVA-v1.6-34B*. The captioning model then re-generated a "refined" description incorporating this feedback. Table 11 shows that the refined captions did not produce major accuracy improvements, suggesting that a single feedback pass is insufficient for significantly enriching image descriptions.

Overall, although the idea of iterative caption refinement has intuitive appeal, our preliminary tests suggest that more elaborate or repeated feedback cycles might be necessary to achieve substantial gains. Even so, the primary finding remains that R1-style reasoning robustly boosts performance relative to standard instruction-tuned reasoning, underscoring the importance of multi-step logic in VQA tasks.

## C  Experiment Details

We thank LLaMA-Factory[6] and EasyR1[7] for open-sourcing the training framework that we used for SFT and GRPO. In Table 13 and 14, we detail the hyperparameters that we used for SFT, GRPO and inference. We further lay out the prompts we used for generating image captions. Experiments were conducted on GPU clusters to the similar level of NVIDIA H100 80GB GPU. SFT/Distillation requires 30 minutes and RL requires 20 hours for each iteration. In addition, distillation data generation with verification requires about 8 hours.

Table 12: Inference hyperparameters.

| | |
|---|---|
| max_new_tokens | 2048 |
| top_p | 0.001 |
| top_k | 1 |
| temperature | 0.01 |
| repetition_penalty | 1.0 |

Table 13: Supervised fine-tuning hyperparameters.

| | |
|---|---|
| Data type | bf16 |
| Learning rate | 5e-7 |
| Global batch size | 32 |
| Scheduler | Cosine |
| Warmup ratio | 0.1 |
| Num train epochs | 1 |
| Image max pixels | 262144 |

**Prompt for image description generation**

Please provide a detailed description of this image that could help another AI model understand it completely. Be specific and comprehensive while maintaining natural language flow.

---

[6]https://github.com/hiyouga/LLaMA-Factory
[7]https://github.com/hiyouga/EasyR1

Table 14: GRPO hyperparameters.

| | |
|---|---|
| Rollout batch size | 512 |
| Global batch size | 128 |
| Max grad norm | 1.0 |
| Data type | bf16 |
| Learning rate | 1e-6 |
| Weight decay | 1e-2 |
| Warmup ratio | 0.0 |
| Rollout temperature | 1.0 |

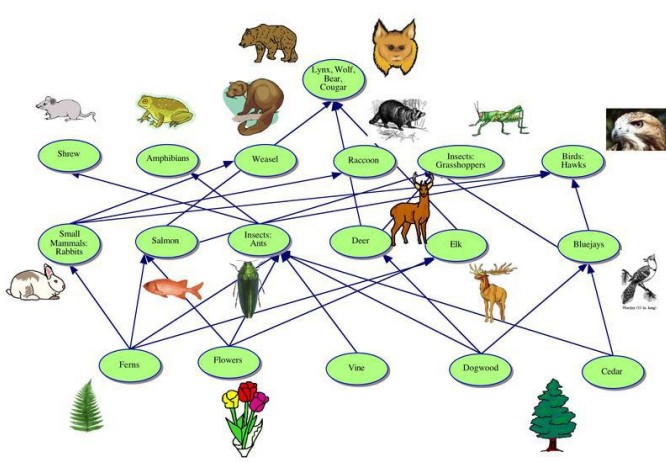

Figure 12: The image to the demonstrated reasoning example in section 1.

## D  Output Examples

**Repetitive reasoning example after SFT without filtering**

But the question is asking for the route and endpoint. So, the route would be the highway number, and the endpoint would be the destination.

Wait, but the signs are arranged vertically. So, the top signs are the routes, and the bottom ones are the destinations. So, if you're at the junction and you're facing the signs, you can see both. So, the route is the highway number, and the endpoint is the destination.

But the question is phrased as "the route and endpoint reached by continuing in a straight line." So, if you're at the junction and you're facing the signs, you can see both the routes and the destinations. So, the route would be the highway number, and the endpoint would be the destination.

Wait, but the instruction is a bit ambiguous. It says "continuing in a straight line," which might mean following the route indicated by the sign. So, if you're at the junction and you're facing the signs, you can see both the routes and the destinations. So, the route is the highway number, and the endpoint is the destination.

But the question is asking for the route and endpoint. So, the route would be the highway number, and the endpoint would be the destination.

Wait, but the signs are arranged vertically. So, the top signs are the routes, and the bottom ones are the destinations. So, if you're at the junction and you're facing the signs, you can see both. So, the route is the highway number, and the endpoint is the destination.

But the question is phrased as "the route and endpoint reached by continuing in a straight line." So, if you're at the junction and you're facing the signs, you can see both the routes and the destinations. So, the route would be the highway number, and the endpoint would be the destination.

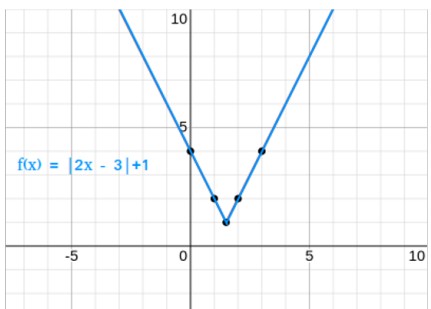

Figure 13: The image to the visual reasoning task that we show for SFT-Iter3 and GRPO-Iter3 output below.

---

**Example of SFT-Iter3 incorrect output and GRPO-Iter3 Correct output to the same question**

Question: The derivative of f(x) at x=2 is _____ that at x=5
Choices: (A) larger than (B) equal to (C) smaller than

**SFT-Iter3**
<think>
Okay, let me see. The question is asking about the derivative of the function f(x) = |2x - 3| + 1 at x=2 compared to x=5. First, I need to recall what the derivative represents. The derivative gives the slope of the tangent line at a point on the function. So I need to look at the slopes at those two points.
</think>

The function ( f(x) = |2x - 3| + 1 ) is an absolute value function, which means it has a V-shape. The vertex of this V-shape occurs where the expression inside the absolute value is zero, i.e., ( 2x - 3 = 0 ), which gives ( x = $\frac{3}{2}$ ). The derivative of ( f(x) ) will be zero at the vertex because the function changes direction there. For ( $x < \frac{3}{2}$ ), the function is decreasing, and for ( $x > \frac{3}{2}$ ), it is increasing. The slopes on either side of the vertex are negative and positive, respectively. At ( x = 2 ) (which is to the right of the vertex), the function is increasing, so the derivative is positive. At ( x = 5 ) (which is further to the right), the function is also increasing, so the derivative is positive but steeper because the function is increasing faster. Since the function is increasing faster at ( x = 5 ) than at ( x = 2 ), the derivative at ( x = 5 ) is larger.
<answer>(C)</answer>
==================
**GRPO-Iter3**
<think>
Okay, let's see. The question is asking about the derivative of the function f(x) = |2x - 3| + 1 at x=2 compared to x=5. First, I need to recall what the derivative represents. The derivative at a point gives the slope of the tangent line at that point. So I need to look at the slopes at x=2 and x=5.
</think>

The function ( f(x) = |2x - 3| + 1 ) is an absolute value function shifted up by 1. The absolute value function ( |2x - 3| ) has a V-shape with a vertex at ( x = $\frac{3}{2}$ ). The derivative of ( |2x - 3| ) is ( 2 ) for ( $x > \frac{3}{2}$ ) and ( -2 ) for ( $x < \frac{3}{2}$ ). Adding 1 to the function doesn't affect the derivative's magnitude but shifts the y-values up. At ( x = 2 ), which is greater than ( $\frac{3}{2}$ ), the derivative is ( 2 ). At ( x = 5 ), which is also greater than ( $\frac{3}{2}$ ), the derivative is also ( 2 ). Since the derivative is the same at both points, the slopes are equal.
<answer>(B)</answer>

Ground truth: (B)

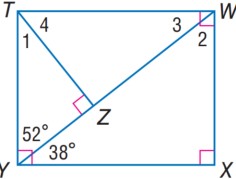

Figure 14: The image to the task that we show for SFT-Iter3 and GRPO-Iter3 output below.

---

**Example of SFT-Iter3 incorrect output and GRPO-Iter3 Correct output to the same question**

Question: Find $m\angle3$.
Choices: (A) 28 (B) 38 (C) 52 (D) 62

**SFT-Iter3**
<think>
Okay, let me see. The problem is asking for the measure of angle 3 in the given quadrilateral. The hint suggests looking at the angles provided, which are 52 degrees and 38 degrees. Hmm, the quadrilateral is a rectangle, so all angles should be 90 degrees. But the hint is confusing me. Let me check the options again. The angles 52 and 38 are given, but they're not the ones I need to find. The question is about angle 3, which is opposite angle 1. Since opposite angles in a rectangle are equal, angle 3 should be the same as angle 1. But angle 1 is not given directly. Let me see, the hint is saying to look at the angles provided. The 52 and 38 are supplementary angles, which means they add up to 180 degrees. But angle 3 is not supplementary to 52 or 38. It's the opposite angle, so it should be the same as angle 1. Since angle 1 is not given, I need to use the options. The only angle that could be the same as angle 1 is the one that's not given directly. The options are A, B, C, D. The only one that's not supplementary to 52 or 38 is 62. So angle 3 must be 62 degrees.
</think>

The problem states that the quadrilateral is a rectangle, and all angles in a rectangle are 90 degrees. The hint provided the angles 52° and 38°, which are supplementary angles (52° + 38° = 90°). However, angle 3 is the opposite angle to angle 1, which means they are equal because opposite angles in a rectangle are congruent. Since angle 1 is not given directly, we need to look at the options provided. The only angle that fits the properties of opposite angles in a rectangle is the one that is not supplementary to the given angles. The options are A (28), B (38), C (52), and D (62). Since angle 1 is not given, the only angle that could be the same as angle 3 is the one that is not supplementary to 52° or 38°. The only angle that fits this criterion is 62°, as it is the only one not listed as supplementary to the given angles.
<answer>(D)</answer>
==================
**GRPO-Iter3**
<think>
Okay, let's see. The question is asking for the measure of angle 3. The image provided is a rectangle with some angles labeled. The hint suggests looking at the angles given, which are 52 degrees and 38 degrees. Hmm, the rectangle has right angles, so each angle should be 90 degrees. The angles 52 and 38 are supplementary because they add up to 90 degrees (90 - 52 = 38). But angle 3 is not one of those. It's the angle opposite the 38-degree angle, so it should be the same. Wait, the question is about angle 3, which is the one opposite the 38-degree angle. So it should be 38 degrees too, right?
</think>

The problem states that angle 3 is opposite the 38-degree angle. In a rectangle, opposite angles are equal. Since angle 3 is opposite the 38-degree angle, angle 3 must also be 38 degrees.
<answer>(B)</answer>

Ground truth: (B)

---

# E   Impact Statement

Our work introduces *OpenVLThinker*, an open-source large vision–language model (LVLM) designed explicitly for complex reasoning tasks through a novel iterative self-improvement process integrating supervised fine-tuning (SFT) and reinforcement learning (RL).

Positive societal impacts include significant improvements in multimodal reasoning, beneficial for applications in education, assistive technologies, accessibility, and content moderation, potentially enhancing decision-making processes involving visual and textual data.

However, improved generative capabilities inherently pose potential negative societal impacts, such as misuse for creating deceptive or manipulative content or misinformation. There are also fairness and privacy considerations, as biases in training data or misinterpretation of visual information could unfairly affect specific groups.

To mitigate these risks, responsible deployment practices such as gated model releases, comprehensive documentation, usage guidelines, and continuous monitoring for misuse and biases are essential.

