# OpenReview forum: "OpenVLThinker: Complex Vision-Language Reasoning via Iterative SFT-RL Cycles"
_NeurIPS.cc/2025/Conference — NeurIPS 2025 poster_

### Official Review · Reviewer_iVRi · 2025-06-28

**Clarity:** 2
**Significance:** 2
**Originality:** 2
**Rating:** 4
**Confidence:** 3

**Summary:**

This paper introduces OpenVLThinker-7B, an open-source Large Vision-Language Model (LVLM) based on Qwen2.5-VL-7B, designed to exhibit complex chain-of-thought (CoT) reasoning. The core methodology is an "iterative SFT-RL cycle", where Supervised Fine-Tuning (SFT) is alternated with Reinforcement Learning (RL) using Group Relative Policy Optimization (GRPO). The authors propose that SFT helps "surface latent reasoning actions", and RL subsequently refines these. In each iteration, the base model is retrained using data generated by the model from the preceding iteration. The paper reports performance gains on six benchmarks, such as +3.2% on MathVista and +2.7% on HallusionBench, claiming these are achieved with high data efficiency (12K samples per iteration) compared to concurrent methods.

**Questions:**

- The paper asserts the Qwen2.5-VL-7B base model "seldom exhibits reasoning behaviors initially". This contrasts with findings from Chen et al. (VLAA-Thinker), who note reflective patterns in the same base model family prior to RL. Could you please discuss this discrepancy?

- VLAA-Thinker also suggests SFT can induce "pseudo reasoning paths," contrasting with your view of SFT effectively surfacing actions and narrowing the RL search space. How do you reconcile these differing perspectives on SFT's role?

- Given that SFT on text-based CoTs (from DeepSeek-R1 and captions) is stated to cause a performance drop due to "a lack of precise visual grounding", could you provide a detailed explanation of the specific mechanism by which GRPO, an RL algorithm optimizing policy based on textual output preferences, directly rectifies this multimodal visual grounding issue?

- "Iterative Self-Improvement" involves retraining from the base Qwen2.5-VL-7B model in each iteration using data from the previous iteration's model. Why is this termed "iterative self-improvement" of a single evolving model rather than an iterative data refinement strategy that produces a sequence of distinct, newly trained model instances?

- What are the advantages of this full retraining approach versus continuously fine-tuning the Iter(i-1) checkpoint to create Iter(i), especially if aiming for "self-improvement"?

**Ethical Concerns:**

["NO or VERY MINOR ethics concerns only"]

**Final Justification:**

The authors successfully addressed all major criticisms with a rebuttal, including crucial new ablation studies that validate their iterative SFT-RL methodology's superiority over single-stage approaches. They clarified the method's mechanistic underpinnings and reconciled their findings with concurrent work, resolving initial concerns about the paper's scientific rigor. The paper's claims are now substantially better supported and its contributions more clearly articulated.

**Limitations:**

yes.

**Quality:**

2

**Strengths And Weaknesses:**

$$\textbf{Strengths}$$

- The paper claims data efficiency using only 12K training samples per iteration to achieve its results, which is considerably less than several concurrent 7B models using single-iteration SFT+RL (e.g., 150K-270K).

- Provides an open-source LVLM (OpenVLThinker-7B) capable of enhanced reasoning is valuable for the research community.

- The process of iteratively generating better training data using the previous iteration's model to train a new model instance from a base checkpoint is systematic. Figure 3 shows progressive improvement on MathVista, supporting this data refinement strategy.

- The attempt to analyze how SFT and RL affect the occurrence of specific keywords (e.g., "wait," "check") is a good step towards understanding model behavior changes, forming the basis of the "action highlighting" hypothesis.

$$\textbf{Weaknesses}$$

- The paper states the model is retrained "from scratch at each iteration" or from the "base model Qwen2.5-VL-7B" using data from the prior iteration. This is more accurately an iterative data refinement and curriculum generation strategy for training new model instances rather than the "self-improvement" of a single, continuously evolving model.

- The SFT "action highlighting" hypothesis, while interesting, is not clearly demonstrated to be mechanistically distinct from SFT providing a good initialization or behavioral prior for RL.

- Some effort could help address conflicting findings from relevant concurrent research. For instance, the paper's premise that the "base model seldom exhibits reasoning behaviors initially" is challenged by Chen et al. (VLAA-Thinker), who report observing reflective reasoning patterns in the same Qwen2.5-VL base model before RL. This discrepancy undermines a key motivation for the proposed SFT role. Similarly, the paper's positive framing of SFT (surfacing latent actions, narrowing RL search space) contrasts with VLAA-Thinker's caution that SFT can induce "pseudo reasoning paths" detrimental to subsequent RL. Given VLAA-Thinker is a cited concurrent model using the same base architecture, this lack of discussion is a major omission.

- The paper argues that initial SFT on text-based CoTs (with image captions) can degrade performance due to "imprecise visual grounding". It then claims the iterative SFT-RL cycle, where RL "refines the model's reasoning skills", rectifies this. Can you provide some clarity on RL's Mechanism for Improving Visual Grounding.

- There is insufficient explanation of how the RL component (GRPO), which primarily optimizes policies based on ranked preferences of textual outputs, directly addresses or improves multimodal visual grounding. This link is critical but underspecified.

---

> ### Author Rebuttal · Authors · 2025-07-31
>
> Thank you for the detailed feedback. Please find our responses below. We hope that our clarifications and additional experiments resolve the concerns and any potential confusions.
>
> ---
>
> ### **W1: Why term the process “iterative self‑improvement” when each cycle trains a fresh model from the base checkpoint? Please justify full restarts versus continuing fine‑tuning and explain any resulting gains.**
>
> Thank you for the suggestion. As clarified in footnote 2 (page 6, line 177) of our manuscript, **restarting training from scratch at each iteration is a standard practice in iterative self-improvement methods [1, 2]**. This design choice ensures training stability and prevents overfitting, thus maintaining better generalization to unseen tasks. Notably, ReST^EM [1] explicitly highlights that re-training from the base model provides comparable task-specific performance and significantly better transfer to held-out tasks compared to continued training (Section 3, page 5, Differences with ReST). We emphasize that, in our iterative re-training approach from the base model, **the data is iteratively refined and improved across iterations**.
>
> To further substantiate this design choice, we conducted an additional experiment comparing our approach of re-training from scratch versus continued training from the previous checkpoint. We evaluated these approaches on three representative benchmarks: MathVista (math reasoning), EMMA (general reasoning), and HallusionBench (perception).
>
> |   | MathVista | EMMA | HallusionBench |
> |---|------|----|------|
> | OpenVLThinker (re-training)   | 71.7 | 25.8 | 70.2  |
> | OpenVLThinker (continue training) | 71.8 |  25.1  | 66.8  |
>
> The results indicate that continued training from the previous iteration leads to noticeable performance degradation on HallusionBench, despite similar performance on MathVista and EMMA. We will emphasize this in our revision by elevating the footnote into the main text and including these additional results.
>
> [1] Beyond human data: Scaling self-training for problem-solving with language models
>
> [2] V-star: Training verifiers for self-taught reasoners
>
> [3] Reinforced self-training (rest) for language modeling
>
> ---
>
> ### **W2: Can you supply evidence that SFT’s “action‑highlighting” effect is mechanistically distinct from a simple good initialization for RL**
>
> We appreciate your comment and realize there might be a misunderstanding. We clarify that our arguments explicitly differentiate between two mechanistically distinct roles that SFT plays:
>
> 1. **Highlighting reasoning actions**: SFT explicitly promotes specific reasoning behaviors, such as reflections and verifications, by emphasizing reasoning-trigger keywords. Crucially, without prior highlighting from SFT or inherent reasoning tendencies from the base model, RL alone does not consistently induce these behaviors spontaneously for VL models. To substantiate this, we analyzed the keyword frequency of the RL-trained MM-Eureka (7B) model across 1,000 MathVista examples: (a) "wait": 0 occurrences, (b) "check": 52 occurrences, (c) "maybe": 0 occurrences. On the contrary, SFT effectively and efficiently influence reasoning behavior patterns, distinct from mere performance benefits.
>
> 2. **Initialization for RL**: Independently, the quality of SFT-generated data directly affects the model’s initial performance, facilitating more effective subsequent RL training. This quality-driven initialization is precisely the motivation behind our iterative training approach, which progressively refines the quality of SFT data while retaining the desired reasoning behaviors.
>
> As demonstrated in Figure 6 of our manuscript, iteration 1’s SFT does not provide superior initialization in terms of accuracy; in fact, its performance is lower than the base model. Thus, the primary contribution of SFT at this iteration is not performance initialization but rather the clear distillation of reasoning behaviors, evidenced by significantly increased frequencies of reflective keywords. These behaviors persist through subsequent RL training, remaining stable while RL enhances overall model performance. Consequently, the RL-enhanced model serves as an improved reasoning demonstration source for the following iteration, retaining desired behaviors while benefiting from RL-driven performance improvements.
>
> ---
>
> ### **W3: Reconcile with VLAA‑Thinker’s reports that (a) the same base model already shows reflective reasoning and (b) SFT can create harmful pseudo‑reasoning paths?**
>
> Thanks for the question. We believe this perceived contradiction also roots from the misunderstanding of our iterative approach as well as our arguments toward the role of SFT. We clarify that our findings **align rather than conflict with concurrent papers**.
>
> (a) We must note that several concurrent works have made similar observations as ours. VL-Rethinker [1] explicitly noted that “complex, deliberate thinking patterns, such as explicit self-correction, **did not consistently emerge** as a direct result of standard RL on VLMs.” This is exactly what motivated their approach in forcing the model to rethink during RL by explicitly appending the keyword “wait”. Similarly, our findings suggest these reflective behaviors are latent and require targeted approaches (like iterative SFT-RL cycles) to reliably surface and refine.
>
> (b) We fully acknowledge VLAA-Thinker's observation regarding harmful pseudo-reasoning paths caused by SFT. Our manuscript explicitly states (lines 47–48) that CoT traces from initial SFT provide demonstrations of reasoning actions but **do not immediately improve LVLM’s accuracy** Additionally, we highlight (line 54) that the initial SFT step "**leads to a performance drop**", which is consistent with VLAA-Thinker. In Appendix C (page 18), we specifically illustrate examples of ineffective, repetitive reasoning behaviors post-SFT. Precisely this harmful initial impact motivated our iterative approach: each RL stage refines these reasoning behaviors, generating **progressively improved data** for subsequent SFT steps.
>
> In short, our iterative SFT-RL methodology explicitly addresses and leverages the phenomena described by VLAA-Thinker, rather than contradicting them.
>
> [1] VL-Rethinker: Incentivizing Self-Reflection of Vision-Language Models with Reinforcement Learning
>
> ---
>
> ### **W4: By what specific mechanism does GRPO repair the imprecise visual grounding introduced by caption‑based CoTs?**
>
> In the paper, we mentioned the imprecise visual grounding in the context of the very first SFT stage, where reasoning traces are derived from text-based LLMs (QwQ-32B) conditioned on captions generated by a vision-language model (Qwen2.5-VL-7B). Since captions inherently lose some visual information and precision, the reasoning traces from QwQ-32B may contain inferences or educated guesses based on incomplete information. For example, consider the reasoning of QwQ-32B below, based solely on captioned inputs:
>
> > Question: What is the main color of the can's top surface?
>
> > QwQ-32B: [think]\nOkay, let's tackle this question. The user is asking for the main color of the can's top surface based on the image provided. First, I need to go through the image carefully to find any shows of colors.\n\nLooking at the image details: *The can's top is shown with a pull-tab mechanism. The surface is on a dark wooden table. The pull-tab is in the center with a metallic ring. The instructions on the tab are shown, but no color is specified for the text.* …... Metals can be silver, but maybe the overall top is a different color.\n[/think]\n\n[answer]Silver[/answer]
>
> GRPO improves upon this by progressively shifting the model away from caption-based inference (that it learns during SFT iteration 1) toward reasoning on direct visual input, as it allows the model explore better reasoning paths with verifiable reward. Consequently, OpenVLThinker after iteration 1 already reasons directly from visual inputs, substantially reducing the potential ambiguity in the reasoning traces it generates as compared to the QwQ generated data. Further iterations continue refining the ”teacher model” for next iteration, systematically improving reasoning demonstrations while retaining beneficial reasoning behaviors demonstrated by QwQ-32B.
>
> We will emphasize this point in our next revision to avoid confusion.

---

> > ### Comment · Reviewer_iVRi · 2025-08-04
> >
> > Thank you to the authors for the detailed and comprehensive rebuttal. I have carefully read all the initial reviews and your thoughtful responses. Based on the clarifications, justifications, and the significant new experimental evidence provided, I will now proceed to finalize my score and recommendation for the paper.

---

> > > ### Author Response · Authors · 2025-08-05
> > >
> > > Thank you for your positive feedback! We are grateful for your recognition of our efforts and that our provided clarifications and new experiments were clear and helpful. Please let us know if there is any further remaining questions.

---

### Official Review · Reviewer_5ZWN · 2025-07-02

**Clarity:** 3
**Significance:** 3
**Originality:** 3
**Rating:** 4
**Confidence:** 4

**Summary:**

This paper introduces OpenVLThinker-7B, one of the first open-source large vision-language models (LVLMs) to demonstrate sophisticated chain-of-thought reasoning capabilities similar to proprietary models like GPT-o1. The authors propose an iterative training methodology that alternates between supervised fine-tuning (SFT) and reinforcement learning (RL) stages. The key insight is that SFT acts as an "action highlighting" mechanism that surfaces latent reasoning behaviors (triggered by keywords like "wait", "check"), which RL can then effectively optimize. Starting from Qwen2.5-VL-7B, the model is trained through multiple iterations using only 12K examples per iteration, achieving competitive performance across six challenging benchmarks.

**Questions:**

What is the total computational cost of the iterative training process compared to standard approaches? Could you provide FLOPs or GPU-hours comparisons?

Can you provide results for single-stage SFT-only and RL-only baselines trained on the same 12K examples? This would clarify the benefit of iteration versus simply using both techniques.

Have you tested the approach on other base models (e.g., LLaVA, InternVL)? Is the success tied to specific properties of Qwen models?

Can you elaborate on the connection to EM algorithms mentioned in line 173? A more formal analysis of why iteration helps could strengthen the paper.

In Table 3, "Vanilla" SFT shows a large performance drop (68.5→57.5). Could you analyze what specific aspects of the distilled reasoning traces cause this degradation?

**Ethical Concerns:**

["NO or VERY MINOR ethics concerns only"]

**Final Justification:**

Thank you for answering my question; I decided to keep my original score.

**Limitations:**

Yes

**Paper Formatting Concerns:**

No major concerns.

**Quality:**

3

**Strengths And Weaknesses:**

Strengths

The iterative SFT-RL approach is innovative and well-motivated. The authors provide compelling analysis (Figure 6) showing how SFT surfaces reasoning keywords that are rarely present in the base model, while RL refines performance without dramatically changing the action distribution.

 Despite using only 12K training examples, OpenVLThinker-7B achieves competitive or superior performance compared to concurrent models using 10x more data (150-270K examples). The 2% average improvement over the base model and 2.7% reduction in hallucinations on HallusionBench are notable.

The paper includes insightful analysis of reasoning behaviors, including keyword frequency analysis and the role of SFT in guiding RL exploration. The curriculum RL approach based on data difficulty is also well-designed.

The paper is generally well-written with clear figures illustrating the methodology and results. The examples in Figure 4 and Appendix C effectively demonstrate the model's reasoning capabilities.

Weaknesses

The paper doesn't discuss the computational requirements of the iterative training process. With RL requiring 20 hours per iteration (mentioned in Appendix B), the total training cost could be substantial. How does it compare to baseline methods? I think the improvements seem marginal considering the great cost.

The paper primarily compares against the base model and concurrent methods. More comparisons with standard vision-language training approaches (e.g., single-stage SFT or RL) would strengthen the evaluation.

The initial SFT data generation process using image captions and QwQ-32B is somewhat ad-hoc. The impact of caption quality on final performance could be better analyzed beyond the brief discussion in Appendix A.

All experiments use models from the Qwen family. It's unclear whether the approach generalizes to other model architectures or if the benefits are specific to Qwen's pretraining. See the paper "ognitive Behaviors that Enable Self-Improving Reasoners, or, Four Habits of Highly Effective STaRs".

---

> ### Author Rebuttal · Authors · 2025-07-31
>
> Thank you very much for your support, careful reading and the constructive feedback that helped us improve our work. Please see our detailed response with additional experiments below.
>
> ### **W1. Discuss the overall computational cost of the iterative SFT → RL loop (e.g., FLOPs or GPU‑hours)**
>
> Thanks for pointing this out. In appendix B, we roughly estimated the training time on 2-4 GPUs. Here, we rigorously detail the GPU hours for the SFT and RL processes below with a 8xH100 (or equivalent) node:
>
> | Stage | GPU Hours    |
> |---------|-------------|
> | SFT   | 0.06 (3 min 30 s) * 8 |
> | GRPO-Medium  | 2.01 (2 hrs 35 s) * 8 |
> | GRPO-Hard  | 4.57 (4 hrs 34 min 26 s) * 8 |
>
> The computational cost of the SFT stage is minimal, primarily due to the small dataset size of 3k examples. For RL with medium-difficulty data, we limit the number of training epochs to further manage resource usage effectively. Although the RL stage using hard data (6k examples) incurs the highest computational cost, our method's overall compute demand remains comparable to contemporary RL-based approaches. Crucially, the preceding SFT and medium RL stages expedite convergence during the final RL (hard) stage, and the relatively smaller dataset size reduces the overall computational burden. Nevertheless, we acknowledge that iterative approaches inherently introduce additional computational overhead, and we will explicitly note this in our revised manuscript and update the more precise GPU hours estimate.
>
> ---
>
> ### **W2. Add single‑stage SFT‑only and RL‑only (trained on the same 12 K examples) to show that iteration, not merely using both techniques, brings the lift**
>
> Thank you for suggesting these critical baselines. We conducted the requested experiments under the following settings:
> - RL-only baseline: Qwen2.5-VL trained solely with GRPO on the entire 12K dataset used in OpenVLThinker.
> - SFT-only baseline: Qwen2.5-VL trained solely with SFT on iteration 2 trajectories generated by OpenVLThinker for the same 12K dataset. Similar to the original setup, we filter out examples where the model fails to reach a correct answer within k=4 samplings.
>
> Both baselines were trained until full convergence, with checkpoints selected based on validation performance. In comparison, OpenVLThinker involves both SFT and RL stages starting from the same Qwen2.5-VL initialization.
>
> |     | MathVista | EMMA | HallusionBench |
> |-----|------|------|-------|
> | RL-Only | 71.3 | 24.5 | 66.8 |
> | SFT-Only | 71.1 | 22.3 | 65.4 |
> | OpenVLThinker | 71.7 | 25.2 | 70.2 |
>
> Training RL-only on the complete 12K dataset required approximately 16 hours to finish using an 8xH100 (or equivalent) node, comparable to the cumulative training time across the three iterations of OpenVLThinker. Results indicate that the RL-only approach outperforms the SFT-only baseline, especially on held-out benchmarks like HallusionBench. Nevertheless, OpenVLThinker achieves superior performance compared to both baselines.
>
> ---
>
> ### **W3: How does the quality of the caption‑based SFT data affect final accuracy, and can you provide a deeper analysis on the caption quality?**
>
> Our training data from QwQ-32B was selected based on final-answer correctness. Thus, the quality of captions directly influenced the quantity and richness of the initial dataset available for the first-stage SFT. Higher-quality captions increased the likelihood that the text-based reasoning model would correctly derive final answers instead of random guessing due to lack of information, thus expanding the pool of useful training samples for iteration 1.
>
> In the original setup, we employed Qwen2.5-VL-7B to generate captions and performed rejection sampling from QwQ-32B with k=4 attempts, discarding instances if none yielded correct reasoning. To provide deeper insights, we conducted additional experiments using Qwen2.5-VL-3B, a weaker captioning model, under identical rejection sampling conditions (k=4). The comparative performance on MathVista is detailed below:
>
> | Iter-1 Caption Type | Qwen2.5-VL | SFT-Iter1 | GRPO-Iter1 | SFT-Iter2 | GRPO-Iter2 | SFT-Iter3 | GRPO-Iter3 |
> |--------------|------------|-----------|------------|-----------|------------|-----------|------------|
> | 3B Caption   | 68.5       | 62.5      | 65.6       | 66.1      | 69.4       | 69        | 70.2       |
> | 7B Caption   | 68.5       | 63.4      | 66.6       | 67.5      | 70.9      | 69.5      | 71.7       |
>
> As demonstrated, better caption quality provides more precise visual grounding at the very first data generation step, enabling the reasoning model to produce more accurate and reliable traces. This in turn improves the initial training dataset and enhances the model's reasoning capabilities across iterations. Eventually, the initial data quality orthogonally complements our iterative training framework, positively influencing final model performance.
>
> ---
>
> ### **W4: Does the method transfer to architectures outside the Qwen family?**
>
> Thank you for raising this important point. To ensure a fair comparison with existing baselines (all trained from Qwen2.5-VL-7B-Instruct), we adhered to the same experimental setting in our paper.  Currently, open-source RL frameworks for vision-language models (EasyR1, OpenRLHF, VeRL) only support models within the Qwen-VL family. Extending these frameworks to other architectures for GRPO is a non-trivial effort beyond the scope of our rebuttal timeframe. To demonstrate that our iteratively evolved dataset maintains its effectiveness beyond the Qwen-based models, we have conducted additional experiments validating the iterative SFT data (distillation) with Gemma-3-4B-IT at iterations 1, 2, and 3.
>
> |     | MathVista |
> |----|-----------|
> | Gemma3-4B   | 50.1 |
> | Gemma3-4B (SFT-Iter1) | 46.1 |
> | Gemma3-4B (SFT-Iter2) | 48.5 |
> | Gemma3-4B (SFT-Iter3) | 50.6 |
>
> We also acknowledge the findings presented in [1] and agree that base-model capabilities significantly influence RL (GRPO) performance due to variations in pre-training quality. Consistent with this idea, we highlight SFT’s crucial role as an intermediate training step (priming) that explicitly surfaces beneficial reasoning behaviors, thereby enhancing subsequent RL effectiveness. We will incorporate this discussion with [1] into our manuscript revision.
>
> [1] Cognitive Behaviors that Enable Self-Improving Reasoners, or, Four Habits of Highly Effective STaRs
>
> ---
>
> ### **Q1: The paper mentions an EM‑style perspective; could you formalize this connection?**
>
> As similar to ReST^EM (an iterative SFT algorithm [1]), our method also decouples data collection (E-step) and policy optimization (M-step) in one iteration. To put this formally,
>
> - Generate (E-step): We generate a SFT dataset $D_i$ at iteration i by sampling reasoning traces from the current policy model $p$:  $𝒚^{i} \sim p_{\theta} (𝒚 |𝒙_{i})$
>
> - Improve (M-step): We leverage the new dataset $D_i$ obtained from the Generate step to improve the policy model p via minimzing the SFT (log-likelihood) loss: $𝐽(\theta) = 𝔼(𝒙,𝒚)∼D_𝑖 [𝑟(𝒙, 𝒚) \log 𝑝_{\theta} ( 𝒚 |𝒙)]$. Additionally, the model is further improved through an unified RL step via the GRPO objective on a fixed dataset $D_{RL}$.
>
> [1] Beyond human data: Scaling self-training for problem-solving with language models
>
> ---
>
> ### **Q2: Table 3 shows a sharp drop for vanilla SFT. What in the distilled reasoning traces causes this degradation, and how might it be mitigated?**
>
> The challenge of small-scale LLMs (<7B) struggling to learn effectively from lengthy chains-of-thought (CoTs) generated by larger models is common to both LVLMs and text-only LLMs. [1] specifically highlights that smaller models benefit more when learning from teachers of a similar scale, rather than substantially larger models. In our observations, repetitive or overly verbose reflections become reinforced during the SFT training, causing looping or redundant behaviors, as demonstrated in Appendix C.
>
> For the vision-language tasks examined in our paper, the degradation is further caused  by the information loss when relying solely on image captions rather than direct visual inputs. Consequently, the teacher model’s reasoning may rely on incomplete or ambiguous information. Even with post-processing, this issue notably impacts performance. For example, QwQ-32B attempts to reason from captions that lack crucial visual details:
>
> > Question: What is the main color of the can's top surface?
>
> > QwQ-32B: [think]\nOkay, let's tackle this question. The user is asking for the main color of the can's top surface based on the image provided. First, I need to go through the image carefully to find any shows of colors.\n\nLooking at the image details: *The can's top is shown with a pull-tab mechanism. The surface is on a dark wooden table. The pull-tab is in the center with a metallic ring. The instructions on the tab are shown, but no color is specified for the text.* …... Metals can be silver, but maybe the overall top is a different color.\n[/think]\n\n[answer]Silver[/answer]
>
> The mitigation to such performance degradation is the exact motivation of our iterative approach. By iteratively refining the teacher model through cycles of SFT and RL, we produce progressively clearer and more precise reasoning demonstrations, which in turn yield higher-quality distillation data in subsequent iterations.

---

> > ### Author Response · Authors · 2025-08-08
> >
> > Dear Reviewer 5ZWN,
> >
> > Thank you again for your support and recognition of our efforts! We are following up on our rebuttal and to inquire if any questions remain before the discussion period ends. We are happy to provide any further clarification if needed.
> >
> > Best,
> >
> > Authors

---

> > > ### Comment · Reviewer_5ZWN · 2025-08-08
> > >
> > > Thanks, you provided GPU hour breakdowns for your method but didn’t include the requested comparison to standard approaches (single-stage SFT or RL baselines), making it impossible to assess the computational cost-benefit tradeoff. The single-stage experiments are helpful but lack statistical significance testing to determine if the differences are meaningful. Additionally, the generalization concern persists - you only tested SFT components on Gemma due to framework limitations, not the full iterative pipeline, which suggests the approach may be architecture-specific. Most critically, the fundamental question of whether the iterative complexity provides sufficient benefit over simpler alternatives remains unanswered without proper baseline cost comparisons. The paper is technically solid with interesting insights, but these core evaluation gaps prevent me from increasing the score.

---

### Official Review · Reviewer_6vDT · 2025-07-03

**Clarity:** 3
**Significance:** 3
**Originality:** 3
**Rating:** 4
**Confidence:** 5

**Summary:**

This paper introduces an open-source Large Vision-Language Model (LVLM), OpenVLThinker. The model adopts an iterative training strategy that alternates between Supervised Fine-Tuning (SFT) and Reinforcement Learning (RL), gradually introducing and enhancing complex Chain-of-Thought (CoT) reasoning capabilities on top of a non-reasoning vision-language base model. Results show that OpenVLThinker achieves certain performance improvements across six benchmark tasks that require mathematical and general reasoning abilities.

**Questions:**

Please see weaknesses for details.

**Ethical Concerns:**

["NO or VERY MINOR ethics concerns only"]

**Final Justification:**

The concerns have been addressed, thus I raise my score.

**Limitations:**

yes

**Quality:**

3

**Strengths And Weaknesses:**

## Strengths：
1、	The study proposes an iterative SFT-RL loop. The resulting model outperforms open-source vision-language models of comparable scale on multiple reasoning benchmarks and achieves performance comparable to proprietary models such as GPT-4o. Furthermore, compared to models trained with SFT or RL alone, the proposed approach attains superior reasoning capabilities while using only about one-tenth of the training data.
2、	The paper investigates the synergy between SFT and RL in acquiring reasoning capabilities. Specifically, the SFT stage helps narrow the RL search space, guiding the model toward reasonable reasoning traces. RL then further optimizes and enhances the model’s performance.
3、	The study also explores the impact of data difficulty during the RL stage. The results show that a progressive two-stage training strategy—starting with medium-difficulty samples followed by hard ones—more effectively enhances the model’s complex reasoning ability than using simple or uniform difficulty data.

## Weaknesses：
1、	How were the representative keywords for inference-time actions selected? For example, why was "maybe" chosen as the representative keyword for "seeking alternatives," and why was "identify" selected, given that its usage frequency significantly decreases after supervised fine-tuning?
2、	Each training iteration restarts from the base model rather than continuing from the intermediate model of the previous iteration. This design choice might be further explained.
3、	The evaluation benchmarks are relatively limited. Expanding the diversity of benchmarks would better demonstrate the model’s general reasoning capabilities.

---

> ### Author Rebuttal · Authors · 2025-07-31
>
> We appreciate your feedback and make clarifications with additional experiments as below to address the raised questions.
>
> ---
>
> ### **W1: How and why were the representative keywords selected.**
>
> We selected keyword candidates by analyzing word cloud visualizations generated from reasoning outputs across various checkpoints, including QwQ-32B. Final representative keywords were chosen based on their frequency in the statistics and further categorized through manual inspection. For instance, we categorized "maybe" as indicative of the action "seeking alternatives" due to its frequent co-occurrence with reflective reasoning patterns, such as "Maybe I should consider...", "Maybe I can consider..." or “Or, maybe, …”. The co-occurence frequency of “Maybe I” is 28.1% for OpenVLThinker at iteration 1 and 7.9% for QwQ-32B at data generation. Similarly, “identify” is frequently associated with “First, I need to identify …”
>
> This manual selection approach follows established practices in recent reasoning-model literature [1-3]. Specifically, keywords such as "wait," "check," and "maybe" are recognized reasoning indicators in studies of text-based reasoning models. Notably, [1] references the keyword "maybe" in Appendix B, while [2,3] similarly identify keyword sets through manual analysis (detailed in their appendices).
>
> Eventually, the effectiveness of particular keywords in triggering specific reasoning behaviors depends on the reasoning model (in our case, QwQ-32B). Therefore, we prioritized the most frequent keywords highlighted by the word cloud analysis. We will clarify this in our revision.
>
> [1] LLMs Can Easily Learn to Reason from Demonstrations Structure, not content, is what matters!
>
> [2] SEAL: Steerable Reasoning Calibration of Large Language Models for Free
>
> [3] Speculative Thinking: Enhancing Small-Model Reasoning with Large Model Guidance at Inference Time
>
> ---
>
> ### **W2: Design choice on restarting training iteration needs explanation**
>
> Thank you for the suggestion. As clarified in footnote 2 (page 6, line 177) of our manuscript, **restarting training from scratch at each iteration is a standard practice in iterative self-improvement methods [1, 2]**. This design choice ensures training stability and prevents overfitting, thus maintaining better generalization to unseen tasks. Notably, ReST^EM [1] explicitly highlights that re-training from the base model provides comparable task-specific performance and significantly better transfer to held-out tasks compared to continued training (Section 3, page 5, Differences with ReST). We emphasize that, in our iterative re-training approach from the base model, **the SFT data is refined and improved across iterations**.
>
> To further substantiate this design choice, we conducted an additional experiment comparing our approach of re-training from scratch versus continued training from the previous checkpoint. We evaluated these approaches on three representative benchmarks: MathVista (math reasoning), EMMA (general reasoning), and HallusionBench (perception reliability).
>
> |   | MathVista | EMMA | HallusionBench |
> |---|------|----|------|
> | OpenVLThinker (re-training)   | 71.7 | 25.8 | 70.2  |
> | OpenVLThinker (continue training) | 71.8 |  25.1  | 66.8  |
>
> The results clearly indicate that continued training from the previous iteration leads to noticeable performance degradation on HallusionBench, despite similar performance on MathVista and EMMA. We will emphasize this in our revision by elevating the footnote into the main text and including these additional results.
>
> [1] Beyond human data: Scaling self-training for problem-solving with language models
>
> [2] V-star: Training verifiers for self-taught reasoners
>
> [3] Reinforced self-training (rest) for language modeling
>
> ---
>
> ### **W3: Evaluation benchmarks are relatively limited**
>
> Thank you for your suggestion. In the paper, **we selected six widely-used vision-language benchmarks covering diverse reasoning aspects, including mathematical reasoning, general reasoning, and perceptual reliability**. These benchmarks are consistently employed in technical reports of proprietary models (e.g., GPT, Gemini) and open-source models (e.g., Qwen-VL, Intern-VL). Our evaluation setup further aligns with concurrent works [1,2] on vision-language reasoning, which similarly employ 5–7 benchmarks with a particular emphasis on mathematical reasoning.
>
> Notably, our evaluation benchmarks are comprehensive and all span multiple evaluation subfields. We reported the overall performance to maintain conciseness. For example, EMMA evaluates Math, Chemistry, Physics, and Code through both multiple-choice and free-form questions. To provide clarity on evaluation coverage, we include detailed subset performances on EMMA below:
>
> |   | EMMA-Math | EMMA-Chemistry | EMMA-Physics | EMMA-Code |
> |-----|------|------|-------|------|
> | Qwen2.5-VL | 24.6   | 21.9   | 29.5 | 28.0  |
> | VLAA-Thinker | 28.1  | 22.3   | 28.8 | 27.3  |
> | OpenVLThinker | 28.8  | 22.6   | 32.7 | 26.8 |
>
> Additionally, in response to your concern, we have expanded our evaluation by including two recent benchmarks: MM-Star and WeMath (both released in 2024). MM-Star is particularly comprehensive, assessing six core LVLM capabilities: fine-grained perception, coarse perception, mathematics, science & technology, logical reasoning, and instance reasoning.
>
> |  | **MMStar** | **WeMath** |
> |---|---|---|
> | Qwen2.5-VL| 53.9 | 61.9 |
> | VLAA-Thinker | 55.4 | 62.4 |
> | OpenVLThinker| **61.9** | **64.1** |
>
> We believe our expanded evaluation across these eight comprehensive benchmarks sufficiently demonstrates the effectiveness and generalizability of our approach.
>
> [1] Vision-R1: Incentivizing Reasoning Capability in Multimodal Large Language Models
>
> [2] SFT or RL? An Early Investigation into Training R1-Like Reasoning Large Vision-Language Models

---

> > ### Comment · Reviewer_6vDT · 2025-08-05
> >
> > The concerns have been addressed, thus I raise my score.

---

> > > ### Author Response · Authors · 2025-08-05
> > >
> > > Thank you for your positive feedback on our rebuttal! We are very glad our responses addressed your concerns.

---

### Note · Authors · 2025-08-12

We sincerely thank all reviewers for their insightful comments and the AC's efforts in handling our submission and encouraging discussion.

For reviewer 6vDT,
- We clarified two key points: our keyword selection process, and that restarting training from scratch at each iteration is a standard practice in iterative self-improvement.
- We clarified our use of benchmarks. To further strengthen our findings, we expanded our evaluation on two additional benchmarks.
- We are delighted that these clarifications fully resolved the reviewer's concerns, and they have confirmed they will raise their score.

For reviewer 5ZWN,
- We thank the reviewer for their positive evaluation of our work as innovative, technically solid, and well-motivated.
- Regarding W2, we trained SFT-only and RL-only models for as long as the loss converges, showing our iterative method outperforms both. In the rebuttal, we reported that the RL-only baseline required approximately 16 hours, which is equivalent on scale to our iterative approach. Although a SFT-only baseline is faster (~20 minutes), it performs worse. Crucially, our approach achieves a higher performance ceiling that the baselines cannot reach, even with prolonged training.
- We are glad that we resolved W3, with additional experiments confirming that higher-quality initial data leads to better final performance.
- Regarding W4, The choice of Qwen-VL ensures direct comparison with concurrent work and was a practical constraint, as open-source RL frameworks for VLMs lack ready support for other architectures. Our experiment improving a Gemma model via SFT was a good-faith effort to show cross-architecture potential of our evolving data.

For reviewer iVRi
- We addressed their primary questions by clarifying that iterative re-training is a standard practice and providing new evidence that justifies our approach.
- We also clarified our work’s position relative to concurrent research and how our method remedies the initial drawbacks of SFT.
- We are grateful the reviewer found our rebuttal detailed and acknowledged our clarifications and the significant new experimental evidence. As there are no follow-up questions, we believe all concerns are resolved.

In summary, we are encouraged by the positive consensus. Our revision will incorporate the new experiments and clarify our methodology and compute costs to reflect the discussion from this review process.

---

### Decision · Program_Chairs · 2025-09-17

**Decision:**

Accept (poster)

**Comment:**

The paper received three expert reviews. The authors provided a rebuttal that attempted to address the concerns raised in the reviews. The reviewers read the rebuttal and engaged with the authors. After the rebuttal and discussion, the reviewers gave final recommendations of borderline accept, borderline accept, and borderline accept.  The reviewers unanimously like the paper and recommended accept. The area chair agreed with the recommendation and decided to accept the paper.

Congratulations! Please see the reviews for feedback on the paper to revise the final version of your paper and include any items promised in your rebuttal.